# Hexanematic crossover in epithelial monolayers depends on cell adhesion and cell density

Julia Eckert [1,4], Benoît Ladoux [2], René-Marc Mège [2], Luca Giomi[3] & Thomas Schmidt [1] ✉

Changes in tissue geometry during developmental processes are associated with collective migration of cells. Recent experimental and numerical results suggest that these changes could leverage on the coexistence of nematic and hexatic orientational order at different length scales. How this multiscale organization is affected by the material properties of the cells and their substrate is presently unknown. In this study, we address these questions in monolayers of Madin-Darby canine kidney cells having various cell densities and molecular repertoires. At small length scales, confluent monolayers are characterized by a prominent hexatic order, independent of the presence of E-cadherin, monolayer density, and underlying substrate stiffness. However, all three properties affect the meso-scale tissue organization. The length scale at which hexatic order transits to nematic order, the "hexanematic" crossover scale, strongly depends on cell-cell adhesions and correlates with monolayer density. Our study demonstrates how epithelial organization is affected by mechanical properties, and provides a robust description of tissue organization during developmental processes.

Various processes in biological tissues – i.e. from the patterning and folding of the embryo during gastrulation, to wound healing and cancer progression – leverage on cells' collective migration[1–5]. The latter is mediated by complexes of adhesion molecules connecting cells between each other and with the extracellular matrix, as well as by cells' geometry and cytoskeletal activity. An extracellular change, for instance, can trigger the recruitment of focal adhesion molecules and the rearrangement of the actin network at the cell-substrate interface and cell cortex, thereby causing actomyosin contractions[6–10]. These, on the other hand, affect the shape of the cells and the forces they exert, thus influencing the structure of the tissue at larger length scales[3,11,12].

Among the various physical aspects of multicellular organization, orientational order is often regarded as instrumental to achieve coordination across length scales, by virtue of its inherent propensity

towards enhancing the coherence of microscopic forces that would be incoherent (randomly oriented) otherwise[13–19]. Because of their predominantly anisotropic shape, cells tend to align with each other when they reach confluency, thereby giving rise to liquid crystal phases collectively known as $p$-atics, where $p$ is an integer reflecting the symmetry of the individual building blocks under rotation by $2\pi/p$. Thus, elongated cells – such as fibroblasts[13], neurons[14], and potentially any mesenchymal phenotypes – have been observed to form polar[20,21] (i.e. $p = 1$) or nematic[14,15,22–24] (i.e. $p = 2$) phases, whose spatial structure and dynamics facilitate a number of biomechanical processes. These include the onset of organism-wide cellular flows during gastrulation[25], the development of protrusion and tentacles-like features[18,26], or the extrusion of apoptotic cells[15]. By contrast, various epithelial phenotypes form honeycomb-like structures held

[1]Physics of Life Processes, Leiden Institute of Physics, Universiteit Leiden, 2333 CC Leiden, The Netherlands. [2]Université Paris Cité, CNRS, Institut Jacques Monod, F-75013 Paris, France. [3]Instituut-Lorentz, Leiden Institute of Physics, Universiteit Leiden, P.O. Box 9506, 2300 RA Leiden, The Netherlands. [4]Present address: Centre for Cell Biology of Chronic Disease, Institute for Molecular Bioscience, The University of Queensland, St. Lucia, Brisbane, QLD 4072, Australia. ✉e-mail: schmidt@physics.leidenuniv.nl

together by cadherin-mediated junctions, characterized by hexatic (i.e. $p = 6$) order[27–29].

More recently, this picture has been further elaborated by experimental[30] and theoretical[31] studies, which indicated that certain epithelial layers feature, in fact, a combination of nematic and hexatic order, with the former being dominant at long and the latter at small length scales. These two types of liquid crystal order crossover at an intermediate length scale, corresponding roughly to clusters of order ten cells for MDCK GII on glass[30]. At this scale – here referred to as hexanematic crossover scale, $R_\times$ – the local hexagonal structure inherited from the shape of individual cells is gradually replaced by the uniaxial arrangement caused by clustering of cells into chain-like assemblies, thus causing the emergence of nematic order at larger length scales (Fig. 1). The specific magnitude of the hexanematic crossover scale and how this compares with other length scales in the system is believed to be functional to the organization of epithelial layers, for instance by dictating the specific migration strategy among the various epithelial cells have at their disposal. In this respect, metastatic cells collectively flowing into micron-sized channels of the extracellular matrix are likely to take advantage of the hexatic 6-fold structure found at small length scales, whereas, e.g., the organism-sized vortical flow observed in Drosophila during the germ-band extension, is more likely to rely on the nematic 2-fold organization of cells large length scales. Yet, what sets the crossover length scale and how it can be controlled by epithelia in order to accomplish their biological functions is presently unknown.

Here, we address these questions experimentally and expand the current understanding of multiscale hexanematic order to include aspects which are hardly accessible to both continuum and discrete theories of tissues, especially those that, like in ref. 31, focus on the mechanical aspects of collective cell behavior. These includes, in particular, the role of adhesion of the cells between each other and with the substrate, which, being mediated by a complex regulatory network of signaling pathways, is *de facto* impossible to be accurately incorporated in a mechanical model. Specifically, we demonstrate that, in MDCK-II WT and E-cad KO layers, the hexanematic crossover scale $R_\times$ is affected by the cell densities and stiffness of the underling substrate. We demonstrate that the hexanematic crossover shifts towards shorter length scales for decreasing monolayer density and reduced cell-cell interaction. Furthermore, we show that this process is phenotype-dependent, and suggest that $R_\times$ provides a phenotypic parameter to discerns whether cellular behavior is individual or collective.

## Results

### Reduced cell-cell adhesion increases the shape index and decreases the monolayer density

The cell density in confluent epithelial monolayers affects the morphology and the motility of cells therein[32,33]. It has been proposed that

changes in cell shape and motility crucially depend on the development of stable cell-cell adhesions, because of their interplay with the cellular contractility[34].

As a starting point, we investigated how the cell shape is affected by the interaction between cells. To this end, we compared the shape of epithelial MDCK type II wild-type (WT) cells with that of MDCK-II E-cadherin knock-out cells (E-cad KO)[19]. A reduced level of cell-cell contacts was maintained in MDCK E-cad KO cells through cadherin-6[19]. Both cell lines, MDCK WT and E-cad KO, were cultured for three days, including two days at confluency, on glass. Subsequently, samples were fixed and immunostained for the tight junction protein ZO-1, which is localized near the apical surface of cells to determine cell boundaries. We thus used the ZO-1 signal to identify the cell vertex positions and reconstructed a polygon of each cell (Fig. 2a, b). Using the polygon, we calculated the shape index $p_0$, defined as the ratio between a cell's perimeter $P$, and the square root of its area $A$, $p_0 = P/\sqrt{A}$[35–37], conventionally used as an indicator of the cell's shape. By averaging over all mean cell shape indices of all monolayers imaged, we identified that MDCK WT cells had a smaller shape index of $4.06 \pm 0.07$ (mean ± s.d.) compared to MDCK E-cad KO cells with $4.20 \pm 0.21$ (mean ± s.d.) (Fig. 2c). This observation is in line with the smaller cell aspect ratio for MDCK WT cells compared to MDCK E-cad KO cells that has been reported earlier[19].

When we compared the morphology of the MDCK WT and E-cad KO cell-monolayers, it appeared that MDCK WT cells were more densely packed and roundish (Fig. 2a), while MDCK E-cad KO cells were larger and elongated (Fig. 2b). We then asked whether the density distributions between both cell lines were different, and whether that was reflected in the shape index. To address this questions, we calculated the mean cell-cell distance between neighboring cells as an indicator for the cell density. By comparing the distributions between both cell lines, we found that MDCK E-cad KO cells, on average, assumed a larger cell-cell distance of $19.9 \pm 1.0$ μm (mean ± s.d.), i.e. one cell per $200 \pm 103$ μm², compared to MDCK WT cells of $16.4 \pm 2.1$ μm (mean ± s.d.), i.e. one cell per $295 \pm 153$ μm², (Fig. 2d). Therefore, it is conceivable that the larger shape index of MDCK E-cad KO cells was caused by the increased cell-cell distance, i.e. decreased monolayer density.

To assess the correlation between the shape index to the monolayer density, we grouped monolayers according to their mean cell-cell distance, $R_{cc}$, in six intervals (D1–D6 in Fig. 2d). Upon increasing the cell-cell distance – thus decreasing the monolayer density – we observed a monotonic increase in the shape index for both cell lines (Fig. 2e), corroborating previous experimental observations[33]. MDCK E-cad KO cells assumed a significant larger shape index compared to MDCK WT cells at smaller comparable density intervals ($R_{cc} \leq 20.1$ μm; D3-D4). For the largest cell-cell distance interval measured ($20.1$ μm $\leq R_{cc} < 22.7$ μm; D5), the mean shape indices for both cell lines were indistinguishable.

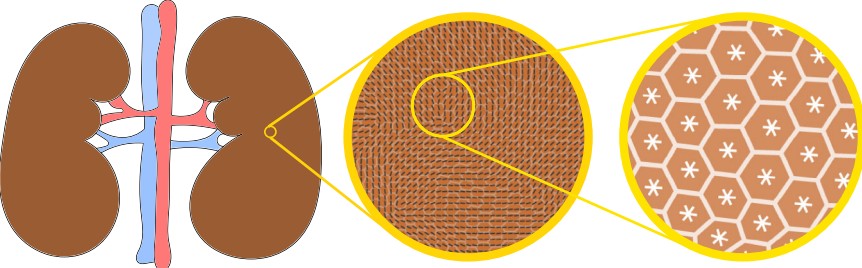

**Fig. 1 | Illustration of multiple shape order in tissues.** Many tissues are dynamical entities that need to adapt their shape over time (left). Adaptation involves collective movement of cells at the tissue-scale, in which cells align with each other and with the direction of tissue flow. This state is faithfully described as a long-range nematic phase indicated by black lines (middle). Yet at increased resolution, at the cell scale, shape order is dominated by an area-filling hexatic phase, dominated by the properties of individual cells (right). As shown throughout this report, cell-specific as well as external parameters control the length scale at which collective behavior toward a nematic phase is observed, that is essential for tissue function and integrity.

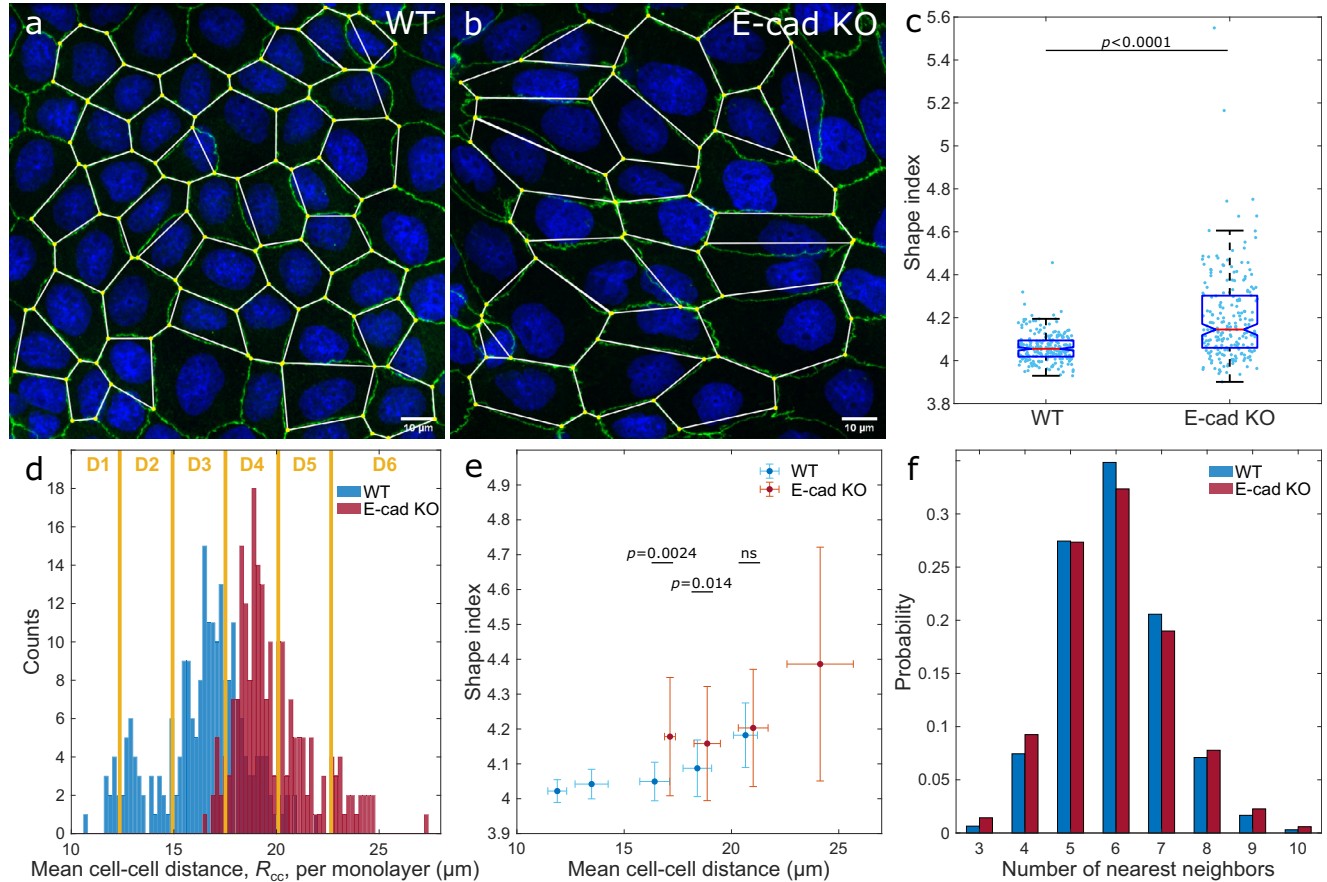

**Fig. 2 | The cell shape index decreases with increasing monolayer density.**
**a**, **b** Confocal image of confluent MDCK-II WT and E-cad KO monolayers (green, ZO-1 and blue, nuclei), cultured on non-coated glass. Cells were segmented, and their shape reconstructed by connecting the vertices. **c** Mean cell shape index per experimental image of both cell lines. The box shows the median (red line), 25th and 75th percentiles (box), maximum and minimum without outliers (whiskers), and 95% confidence interval of the median (notches). **d** Distribution of the mean cell-cell distance, $R_{cc}$. The bin-width is 0.2 μm. For further analysis, monolayers were grouped in six intervals, D1–D6, of 2.6 μm-width each. **e** Mean shape index increases with $R_{cc}$. Error bars represent the standard deviation. **f** The probability distribution of the number of nearest neighbors per cell. Statistics: **c**–**f** $N_{WT} = 226$ and $N_{KO} = 216$ from three independent experiments. **c**, **e** $p$-values are calculated from two-sided Wilcoxon rank sum test: $^{ns}p > 0.5$. Source data are provided as Source Data file.

Taken together, MDCK E-cad KO cells had an overall larger shape index and a smaller cell density compared to MDCK WT cells. In the same density interval, MDCK E-cad KO cells had either a larger or the same shape index compared to MDCK WT cells.

## Hexagonality in epithelia increases with density

As previously mentioned, epithelial layers exhibit hexatic order at the small scale, by virtue of the approximate 6-fold symmetry of individual cells. The latter, in turn, is a natural consequence of the fact that, in two dimensions, isotropic particles pack more densely when arranged in a honeycomb lattice. For rigid disks, this mechanism yields the packing fraction $\phi_{honeycomb} = \pi\sqrt{3}/6 \approx 0.91$, while the remaining fraction, i.e. $1 - \phi_{honeycomb} \approx 0.09$, is occupied by the gaps in between the disks. Evidently, the same limitation does not exist in the case of deformable particles, as these can fill the gaps by adapting to the hexagonal geometry of their neighborhood, eventually reaching confluency (i.e. $\phi_{confluent} = 1$). These considerations suggest that the hexagonality of individual epithelial cells would increase with the monolayer density.

To test this hypothesis, we measured the 6-fold shape function, $\gamma_6$, introduced in ref. 30 in both, the MDCK WT and the E-cad KO cell line, and compared it with the 2-fold shape function, $\gamma_2$, for all individual cells (Fig. 3a and Supplementary Fig. S1a, b). These two functions are examples of a generic $p$-fold shape function, defined in Eq.(1), quantifying the resemblance of a cell, or equivalently any arbitrary polygon, to a $p$-sided regular polygon (or rod for $p = 2$), having the same position

and size. Subsequently, we calculated the ensemble average of the shape functions, $\langle|\gamma_p|\rangle$, (see Eq. (2) in Methods). Regardless of the monolayer density and cell line, the mean 6-fold shape function was found to be always larger than the mean 2-fold shape function at the scale of individual cells (Fig. 3b; Supplementary Fig. S1a, b). Yet we noted that, upon increasing cell-cell distance, the difference $\langle|\gamma_6|\rangle - \langle|\gamma_2|\rangle$ decreased (Fig. 3c). At the largest cell-cell distance interval (D6) of $24.1 \pm 1.5$ μm (mean ± s.d.), both shapes were equally prominent, while the difference between MDCK WT and E-cad KO cells disappeared (Fig. 3b).

By comparing the individual shape functions of both MDCK WT and E-cad KO cells in the same monolayer density interval, it appeared that the 6-fold shape function did not differ in all overlapping density intervals (Fig. 3b; $p$-value > 0.05). On the other hand, the 2-fold shape function of MDCK E-cad KO cells were significantly larger for cell-cell distances in the interval $R_{cc} \leq 20.1$ μm (D3-D4; $p$-value < 0.0001) and equal in the interval $20.1$ μm $\leq R_{cc} < 22.7$ μm (D5; $p$-value > 0.05) compared to MDCK WT cells. This trend in the behavior of $\langle|\gamma_2|\rangle$ in MDCK WT and E-cad KO cells at similar density intervals echos that observed in the behavior of the shape index (Fig. 2e) as also demonstrated by the large correlation between the two functions (correlation coefficient: $0.93 \leq r \leq 0.97$; see Fig. 3d).

From these observations, we concluded that decreasing monolayer density – increasing cell-cell distance – as well as reducing cell-cell adhesions led to an elongation of cells, which can be equivalently

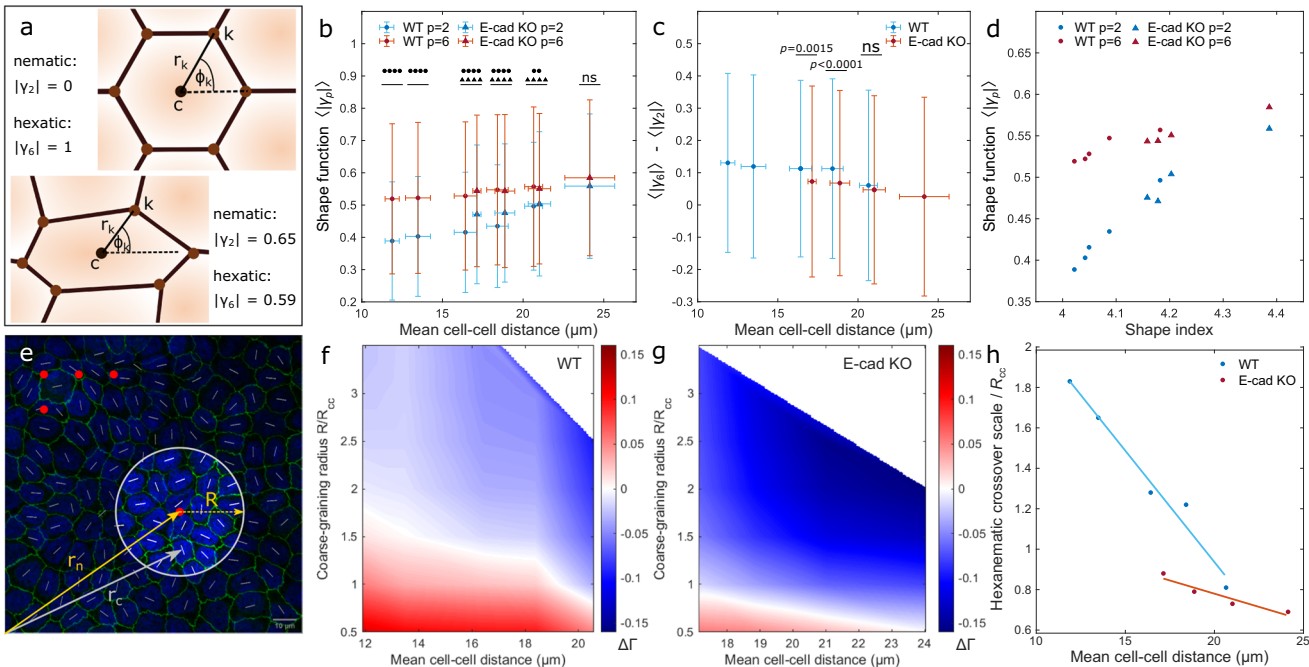

**Fig. 3 | The hexanematic crossover scale depends on the monolayer density and cell line. a** Illustration of the shape function $\gamma_p$ defined in Eq. (1). The magnitude $|\gamma_p|$ of this complex function quantifies the resemblance of the cell shape from that of a regular $p$-sided polygon, while its phase $\mathrm{Arg}(\gamma_p)/p$ yields the cell's $p$-fold orientation. In this example, a perfectly hexagonal cell (top) has $|\gamma_6| = 1$ and $|\gamma_2| = 0$. Conversely, for an elongated hexagonal cell (bottom), both $|\gamma_2|$ and $|\gamma_6|$ are finite. **b** The magnitude of the shape function, $\langle|\gamma_p|\rangle$, for different mean cell-cell distances. Differences between the hexatic and nematic shape functions, $\langle|\gamma_6|\rangle - \langle|\gamma_2|\rangle$, are shown in **c**. The $p$-values for the comparison of different density ranges are presented in Supplementary Table S2. **d** Correlation between the mean shape index and mean shape function. **e** Schematic representation of the coarse-graining procedure used for the computation of the scale-dependent shape parameter $\Gamma_p = \Gamma_p(R)$ for the specific case $p = 2$. To compute the parameter at the point $r_n$, the shape function $\gamma_2$ is averaged within a disk of radius $R$ from $r_n$. The procedure is repeated for each of

the red point, corresponding to the nodes of a regular square grid of lattice spacing $R_{cc}/2$. See the subsection Shape Parameter in Methods for further details. **f, g** The difference between the scale-dependent hexatic and nematic shape parameter, $\Delta\Gamma = \Gamma_6 - \Gamma_2$, plotted as a function of the coarse-graining radius and the mean cell-cell distance for MDCK-II WT (**f**) and E-cad KO cells (**g**). Red and blue tones indicate regions of the parameter space where hexatic order prevails over nematic order and vice versa. The white line marks the hexanematic crossover, which is the length scale $R_\times$ where $\Gamma_6 = \Gamma_2$. Individual plots can be seen in Supplementary Fig. S1c, d. **h** Hexanematic crossover scale versus the mean cell-cell distance. Statistics: **b, c** D1: $N_{WT} = 1266$; D2: $N_{WT} = 2754$; D3: $N_{WT} = 5420$ and $N_{KO} = 572$; D4: $N_{WT} = 2315$ and $N_{KO} = 1480$; D5: $N_{WT} = 232$ and $N_{KO} = 1480$; D6: $N_{KO} = 494$ from three independent experiments. Error bars represent the standard deviation. Two-sided Wilcoxon rank sum test: **$p < 0.01$, ****$p < 0.0001$, ns$p > 0.5$. Source data are provided as Source Data file.

captured by either the cell shape index or the 2-fold shape function, $\gamma_2$. By contrast, 6-fold symmetry always overweights 2-fold symmetry at the single-cell level, independently on the monolayer density and the strength of cell-cell adhesion. Hence, individual cells in confluent monolayers were statistically more hexagonal rather than elongated.

### The absence of E-cadherin shifts the hexanematic crossover towards small length scales

As we anticipated in the introduction, the 6-fold symmetry characterizing the structure of the cellular network at the length scale of individual cells propagates towards larger length scales, giving rise to hexatic order of the monolayer. The hexatic order decays with distance and is eventually replaced by a similarly decaying nematic order at length scales larger than a system-dependent hexanematic crossover scale, $R_\times$. To understand how the crossover lengthscale depends upon the monolayer mechanical and biochemical properties, we coarse-grained the shape functions, $\gamma_2$ and $\gamma_6$, over a disk of radius $R$, thereby obtaining the lengthscale-dependent shape parameter $\Gamma_p = \Gamma_p(R)$ (Fig. 3e and Supplementary Fig. S1c, d; see Methods for details). In our analysis, $R$ was normalized to the mean cell-cell distance, $R_{cc}$. We then analyzed the behavior of the difference $|\Gamma_6| - |\Gamma_2|$ as a function of the coarse-graining radius and the mean cell-cell distance (Fig. 3f–g). In this plot, positive (in red) and negative (in blue) $|\Gamma_6| - |\Gamma_2|$ values correspond respectively to regimes where hexatic order overweights nematic order and vice versa, whereas the white dots mark the hexanematic crossover, where $|\Gamma_6| = |\Gamma_2|$.

In both MDCK WT (Fig. 3f) and E-cad KO cells (Fig. 3g), the hexanematic crossover lengthscale shifted toward smaller and smaller scales upon increasing the cell-cell distance, indicating an increase in the range of hexatic order with the monolayer density. To further highlight this trend, in Fig. 3h we plotted the normalized crossover scale $R_\times/R_{cc}$ for each mean cell-cell distance interval. In both cell lines, $R_\times/R_{cc}$ decreases approximately linearly with the monolayer density, but with significantly different rate. Specifically, at any given monolayer density, the length scale at which the hexatic order prevailed was more significantly reduced for MDCK E-cad KO cells in comparison to MDCK WT cells. Importantly, we note that the difference between the two cell lines is much more prominent when analyzed in terms of the density dependence of the hexanematic crossover scale (Fig. 3h), than in terms of the shape index $p_0$ (Fig. 2e).

Taken together, our results indicate that the range of hexatic order is larger in MDCK WT cells compared to MDCK E-cad KO cells and increases with the monolayer density. Interestingly, we found that the hexanematic crossover scale provides what appears to be a robust indicator to distinguish the two cell lines, thus indirectly conveys information about the nature of the cell-cell interactions, which only for WT cells is mediated by E-cadherin.

### Hexatic order strengthen with the monolayer density
As in molecular liquid crystals, orientational order can be locally disrupted by topological defects, point-like singularities where the cells' local orientation is undefined. In multicellular systems, topological

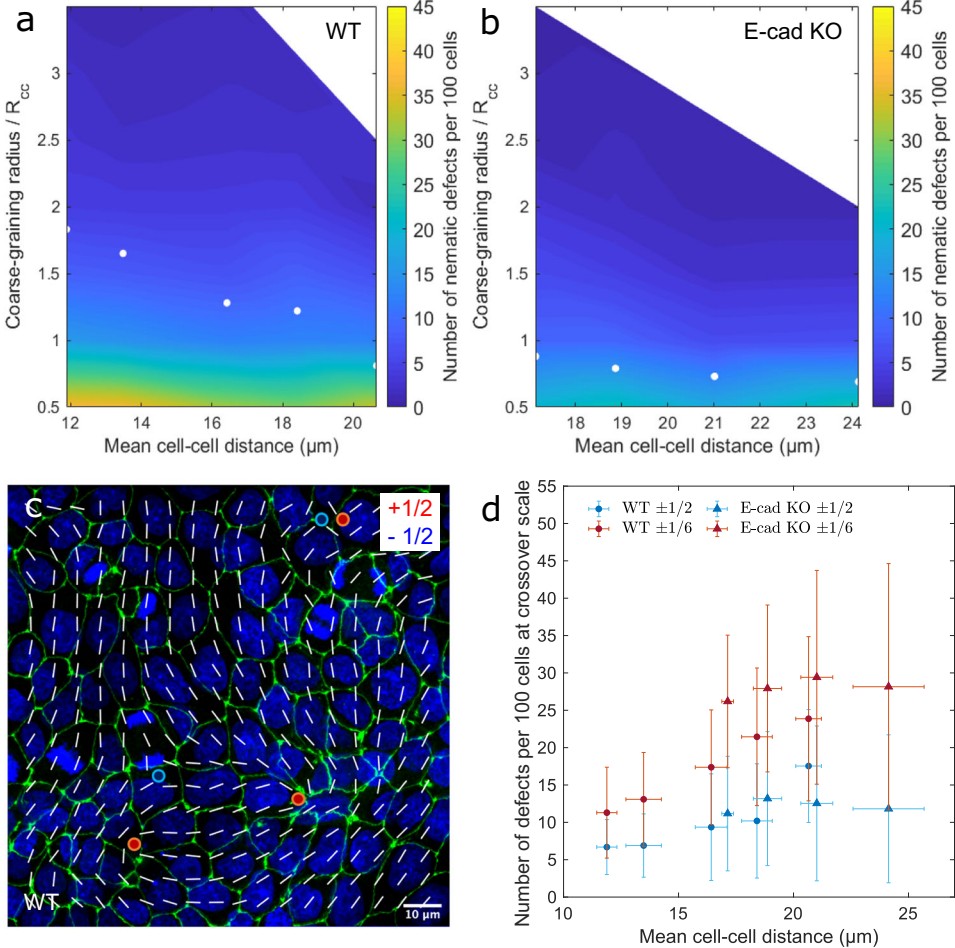

**Fig. 4 | With equal hexatic and nematic order, the defect density per cell depends on the monolayer density. a**, **b** The nematic defect density, defined by the number of defects per 100 cells, as a function of the coarse-graining radius and the mean cell-cell distance for MDCK-II WT and E-cad KO cells, respectively. The number of nematic defects created by the same number of cells was found to be independent of the cell density. At the hexanematic crossover, here marked by white dots, the number of defects increases with increasing mean cell-cell distance.

**c** A smooth nematic director field at the hexanematic crossover with defects of charge $\pm 1/2$ (green, ZO-1 and blue, nuclei). **d** The number of nematic and hexatic defects formed by MDCK-II WT and E-cad KO cells versus the mean cell-cell distance. Statistics: D1: $N_{WT} = 12$; D2: $N_{WT} = 35$; D3: $N_{WT} = 110$ and $N_{KO} = 13$; D4: $N_{WT} = 61$ and $N_{KO} = 123$; D5: $N_{WT} = 8$ and $N_{KO} = 54$; D6: $N_{KO} = 26$ from three independent experiments. Error bars represent the standard deviation. Source data are provided as Source Data file.

defects are believed to serve various biological functionalities, from driving collective motion at length scales significantly larger than that of individual cells[14,24], to facilitate the extrusion of apoptotic cells[15], and the development of sharp features, such as tentacles and protrusions[18,26].

To shed light on the occurrence of topological defects in epithelial cell layers, we computed the nematic and hexatic orientation fields of MDCK WT and E-cad KO cells and determined the location of the corresponding elementary defects (see Methods for details about the defect tracking method). For each cell-cell distance interval and coarse-graining radius, we then computed the corresponding defect density, defined as the number of defects per 100 cells in a monolayer, independently of the cell size. This analysis is shown in Fig. 4a, b, where the density of nematic defects in MDCK WT and E-cad KO cells, respectively, is plotted against the coarse-graining radius and the mean cell-cell distance. Because the smoothing of the orientation field progressively neutralizes pairs of defects and anti-defects, the overall defect density naturally decreases upon coarse-graining (Supplementary Fig. S2a, b). Surprisingly, however, the defect density appeared unaffected by the cell-cell distance (Supplementary Fig. S2c, d for hexatic). In other words, the same number of cells features the same number of defects, independently of their density.

Since nematic and hexatic order occurs in epithelial layers at different length scales, we next investigated how the abundance of

nematic and hexatic defects at the crossover scale, where both types of orientational order are simultaneously present. The abundance is shown in Fig. 4a, b, where the white dots mark the location of the hexanematic crossover for increasing cell-cell distance. Upon computing the number of defects at the crossover scale (Fig. 4c, d), we then found that for MDCK WT cells, the nematic and hexatic defects are more abundant in loosely packed monolayers (large cell-cell distance) (Supplementary Table S3). Consistently with the results summarized in the previous sections, this trend was more significant in MDCK WT than in MDCK E-cad KO cells (Supplementary Table S3), and consistent with the observation that varying the density of MDCK E-cad KO cells has a limited effect on the hexanematic crossover (Fig. 3h).

In conclusion, the defect density, when analyzed at the relevant crossover length scale, was lower for compact monolayers, increasing with decreasing cell density. Given this finding was observed for both shape parameter suggests that both the nematic and the hexatic order together control the collective organization of cells in generating topological defects.

**Lower substrate stiffnesses reinforce the length scale of the hexatic order driven by cell-matrix and cell-cell adhesions**
In the analysis reported so far, we investigated how the biomechanical properties of the cells – their density and mutual adhesion – affect the

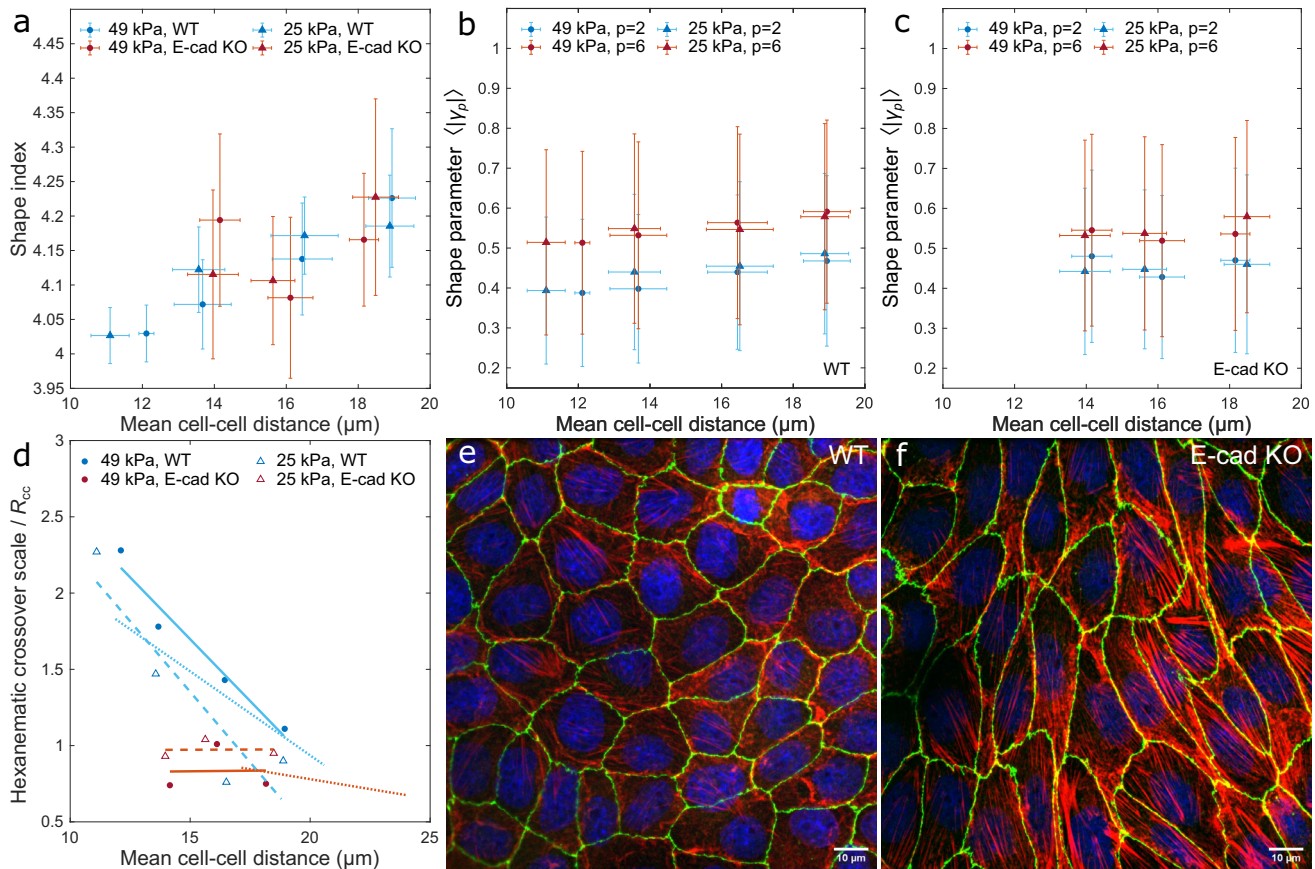

**Fig. 5 | Substrate stiffness has a minor effect on the hexanematic organization.** MDCK-II WT and E-cad KO cells were cultured on PAA gels with stiffness of 49 kPa and 25 kPa. **a** The mean shape index increases with the mean cell-cell distance, $R_{cc}$. Statistics: 25 kPa substrate: D1: $N_{WT} = 62$; D2: $N_{WT} = 13$ and $N_{KO} = 54$; D3: $N_{WT} = 6$ and $N_{KO} = 23$; D4: $N_{WT} = 12$ and $N_{KO} = 4$; 49 kPa substrate: D1: $N_{WT} = 11$; D2: $N_{WT} = 20$ and $N_{KO} = 48$; D3: $N_{WT} = 20$ and $N_{KO} = 91$; D4: $N_{WT} = 21$ and $N_{KO} = 5$ from three to five independent experiments. Error bars represent the standard deviation. **b**, **c** On single-cell scale, the mean hexatic shape function, $\langle |\gamma_6| \rangle$, is always larger compared to $\langle |\gamma_2| \rangle$. Statistics: 25 kPa substrate: D1: $N_{WT} = 7603$; D2: $N_{WT} = 1025$ and $N_{KO} = 3976$; D3: $N_{WT} = 282$ and $N_{KO} = 1282$; D4: $N_{WT} = 415$ and $N_{KO} = 154$; 49 kPa substrate: D1: $N_{WT} = 1108$; D2: $N_{WT} = 1554$ and $N_{KO} = 3480$; D3: $N_{WT} = 985$ and $N_{KO} = 4667$; D4:

$N_{WT} = 753$ and $N_{KO} = 201$ from three to five independent experiments. Error bars represent the standard deviation. **d** The hexanematic crossover scale versus the mean cell-cell distance. Independent of PAA stiffness, MDCK-II WT cells exhibit a stronger dependence on monolayer density compared to E-cad KO cells, as demonstrated by the larger slope of the data. Dotted lines represent the fits of Fig. 3h for cells on non-coated glass. **e** MDCK-II WT cells cultured on non-coated glass appear more isotropic and have a less prominent actin stress fiber network. **f** MDCK-II E-cad KO cells cultured on non-coated glass are stretched and show strong F-actin fibers (red, F-actin, green, ZO-1, and blue, nuclei). F-actin co-staining of WT and E-cad KO cells yielded similar results with a total of 146 and 109 images, respectively. Source data are provided as Source Data file.

hexanematic crossover. Next, we focused on the biomechanical properties of the substrate, in particular its stiffness and adhesion with the cell monolayer. To this end, we cultured both cell lines for two days on fibronectin-coated polyacrylamide (PAA) gels of varying stiffness.

On stiff glass substrates, our measurements of the shape index, $p_0$, (Fig. 2e) and the shape functions, $\langle |\gamma_2| \rangle$ and $\langle |\gamma_6| \rangle$, (Fig. 3h), revealed a dependence of the cell shape on monolayer density. The same trend was found on compliant PAA substrates, whose stiffness ranged between 25 kPa and 49 kPa, but with no evidence of a direct cross-talk between the shape of cells and the stiffness of the substrate. While plated on compliant substrates, cells systematically formed denser monolayers, as demonstrated by the probability distribution of the cell-cell distance (Supplementary Fig. S3). We then asked whether this change in the monolayer density affected the shape index of cells. For that, we grouped monolayers in the six cell-cell distance intervals (D1–D6; Supplementary Fig. S3b–c), as above. The shape index of MDCK WT and E-cad KO cells increased with increasing cell-cell distance. However, most of the density intervals showed no significant indication for a stiffness-dependent shape index: by comparing cells on both PAA gels and glass (Fig. 5a; Supplementary Table S4), data from MDCK WT and E-cad KO cells overlapped. Consistently with previously reported measurements on epithelial monolayers at

jamming[33], our results suggest that the influence of the monolayer density on the shape index overweights that caused by the substrate stiffness, at least for cell-cell distance intervals between 9.7 μm and 28.5 μm and substrate stiffnesses between 25 kPa and 49 kPa.

We next focused on the stiffness-dependence of hexanematic order across length scales. At the scale of individual cells, measurements of the 2-fold and 6-fold shape functions, $\langle |\gamma_2| \rangle$ and $\langle |\gamma_6| \rangle$, demonstrated a prevalence of 6-fold symmetry irrespective of the cell line, PAA gel stiffness, and monolayer density (Fig. 5b, c). On the other hand, comparing the magnitude of the coarse-grained shape parameters, $\Gamma_2$ and $\Gamma_6$, revealed a density-dependence of the hexanematic crossover scale in MDCK WT cells, but not in MDCK E-cad KO cells (Fig. 5d). Together with the fact that MDCK E-cad KO cells feature higher substrate adhesions and stronger actin stress fibers (Fig. 5e, f), reduced intercellular tension, and a lower correlation length in their collective behavior[19], our results suggest that hexatic order could play a role in epithelial phenotypes regardless of the specific properties of the environment.

## Discussion
In this study, we explored the multiscale structure of in vitro layers of MDCK-II cells, with the goal of unveiling how the mechanical and

biochemical properties of the monolayers and the substrate affect the crossover between hexatic and nematic orientational order. Such an example of multiscale organization in living matter has been recently identified by means of numerical simulations and experiments[30,31], and has been conjectured to contribute to the biophysical versatility of epithelial tissues, whose tasks range from organism-wide morphogenetic migration to collective metastatic invasion under strong confinement.

By comparing the behavior of two different cell lines – MDCK WT and MDCK E-cad KO cells – we showed that the existence of hexatic order crucially relies on E-cadherin-mediated intercellular adhesion, whose lack on MDCK E-cad KO cells rendered the cellular shape significantly elongated, hence prone to form nematic phases. Furthermore, as the lower cell-cell adhesion increases the presence of focal adhesions and actin stress fibers[19] and these contribute to the elongation of the cell[38], we suggest that intercellular- and cell-substrate adhesions jointly control the order of cells in monolayers (see also ref. 39). Accordingly, cell-cell adhesion leads to a compact hexagonal shape, whereas actin stress fibers contribute to cell elongation and 2-fold symmetry.

Upon coarse-graining the 2-fold and 6-fold shape functions, we then identified a significant dependence of the hexanematic crossover on the density of the cell monolayer as well as on the specific cell-line. In MDCK-WT cells, in particular, the hexanematic crossover occurs at larger length scales compared to MDCK E-cad KO cells and the 6-fold symmetry inherited by the shape of individual cells persists up to clusters consisting approximately 3–13 cells, depending on the monolayer density. Interestingly, the rate at which the crossover scale increases with density is different for the two cell lines, thus demonstrating a correlation between the *active* mechanisms governing the mechanics of the cell monolayer – i.e. the cell-cell and cell-substrate adhesive interactions – and the structure of the cell monolayers across length scales. These observations, together with previous experimental evidence that increasing density strengthens intercellular adhesion[40,41] while suppressing alignment among stress fibers[40], suggests that, in epithelial layers, multiple physical and biochemical mechanisms could conspire toward consolidating hexatic order at small length scales. On the other hand, our analysis shows that cell density and cell-cell adhesion are efficient control parameters to manipulate the scale of hexanematic crossover, thus it could possibly be used by the system to switch from a hexatic- to a nematic-based migration mode.

Finally, our data provides a clear demonstration that the specific density-dependence of the hexanematic crossover is not universal, but strongly depends on the cells' molecular repertoire and could be used, in principle, to discern among phenotypes along the epithelial-mesenchymal spectrum.

## Methods

### Cell culture
MDCK-II WT (ATCC CCL-34) cells and MDCK-II E-cadherin KO cells were cultured in DMEM(1x)+DlutaMax-1 (2340251, Gibcon) supplemented with 10% foetal bovine serum (FBS; Life Technologies) and 1% penicillin/streptomycin (Life Technologies) at 37 °C with 5% $CO_2$.

### Preparation of polyacrylamide gel substrates
Polyacrylamide (PAA) gels were prepared from a stock solution of 40% acrylamide: A (1610140, Bio Rad) and 2% bis-acrylamide: B (1610142, Bio Rad) in PBS. The ratio of these components were mixed according to the PAA gel stiffness of 25.0 ± 2.4 kPa (6/10 total stock volume A, 15/100 total stock volume B) and 49 ± 8 kPa (6/10 total stock volume A, 28/100 total stock volume B), based on the protocol in ref. 42. PAA gel stiffnesses were measured using indentation-type atomic force microscopy (Chiaro, Optics11 Life) and interpreted based on the Hertz model. The final PAA solution contained 50/100 total volume stock solution, 5/1000 total volume 10% ammonium persulfate (A3678-25G,

Sigma Aldrich) and 15/10,000 total volume TEMED (17919, Thermo Scientific) in PBS. Glass coverslips were plasma activated and incubated with 20 μg ml⁻¹ fibronectin (FC010-10MG, Sigma-Aldrich) for 2 h. After incubation, coverslips were rinsed in water to remove excess protein. Simultaneously, a second set of glass coverslips was silanized. Glass coverslips were plasma activated and incubated with a solution of 2% (v/v) 3-(trimethoxysilyl)propyl methacrylate (440159-500ML, Sigma Aldrich) and 1% (v/v) acetic acid (20104.298, VWR Chemicals) in absolute ethanol for 10 min. After rinsing with 96% ethanol, silanized coverslips were heated at 120 °C for 1 h. PAA gels were sandwiched between the fibronectin-coated glass coverslip and silanized coverslip for 20 min. After polymerization, coverslips were separated and samples were kept in water until cell seeding.

### Immunostaining
Cells on non-coated coverslips and PAA gels were cultured for three days and two days, respectively, reaching confluence after one to two days. After cell fixation with 4% paraformaldehyde (43368; Alfa Aesar) for 15 min, cells were permeabilized with 0.5% Triton-X 100 for 10 min, blocked with 1% BSA in PBS for 1 h. ZO-1 was visualized with anti-ZO-1 rat monoclonal antibody (1:200 ratio; clone R40.76, MilliporeSigma) followed by staining with Alexa Fluor 488 goat anti-rat (1:200 ratio; A11006, LifeTechnology), F-actin with Alexa Fluor 568 Phalloidin (1:200 ratio; A12380, Invitrogen), and DNA with Hoechst (1:10,000 ratio; 33342 Thermo Fischer).

### Imaging
Before imaging, samples were mounted on ProLong (P36962, Invitrogen). Imaging was performed on a microscope setup based on an inverted Axio Observer.Z1 microscope (Zeiss), a Yokogawa CSU-X1 spinning disk, and a 2 ORCA Fusion camera (Hamamatsu). ZEN 2 acquisition software was used for setup-control and data acquisition. Illumination was performed using different lasers (405 nm, 488 nm, 561 nm). Cells were inspected with a 63 × 1.4 oil immersion objective (Zeiss). Images were taken in z-stack focal-planes with distances of 500 nm for a maximal intensity projection.

### Cell shape analysis
**Segmentation.** Cell boundaries of confluent monolayers were analyzed using a maximum intensity projection of z-stack images. Cell segmentation and vertex analysis were performed using custom MATLAB scripts (Mathworks, MATLAB R2018a). In short, the ZO-1 signal was thresholded and skeletonized. Branching points shared by at least three cells were identified as vertices. The number of vertices surrounding a cell corresponds to the number of nearest neighbors. To obtain the polygon structure of each cell, vertices were connected by straight lines.

**Shape function.** The shape function of an arbitrary V-sided polygon, is described by means of the following complex function[30]:

$$\gamma_p = \frac{\sum_{v=1}^{V} |\mathbf{r}_v|^p e^{ip\phi_v}}{\sum_{v=1}^{V} |\mathbf{r}_v|^p}, \tag{1}$$

where $\mathbf{r}_v$ is the position of the $v$-th vertex along the polygon's contour with respect to the polygon's center of mass and $\phi_v = \arctan(y_v/x_v)$ its the angular coordinate. The magnitude $|\gamma_p|$ of the shape function quantifies the resemblance between the cell and a regular $p$-sided polygon having the same size and position, while the phase $\text{Arg}(\gamma_p)/p = \arctan[\Im(\gamma_p)/\Re(\gamma_p)]/p$, with $\Re(\cdots)$ and $\Im(\cdots)$ respectively the real and imaginary part, sets the $p$-fold orientation of the cell. An example of the calculation of $\gamma_p$ is shown in Fig. 3a. For a regular hexagon $|\gamma_2| = 0$, while $|\gamma_6| = 1$ and the 6-fold star representing the phase of $\gamma_6$ is oriented so that the six legs are parallel to the bisectrices of the internal angles. Conversely, for an elongated hexagon, $|\gamma_2|$ and

$|\gamma_6|$ are both finite and, in addition to the 6-fold orientation, a 2-fold orientation can also be identified.

**Ensemble average.** The ensemble average $\langle|\gamma_p|\rangle$ was obtained by averaging the magnitude of the complex function $\gamma_p$ over the entire ensemble of cells, $N_{cell}$, analyzed in a given dataset and density interval. That is

$$\langle|\gamma_p|\rangle = \frac{1}{N_{cell}}\sum_{c=1}^{N_{cell}}|\gamma_p(\boldsymbol{r}_c)|, \qquad (2)$$

where $\boldsymbol{r}_c$ denotes the position of the $c$-th cell, with $c = 1, 2\ldots N_{cell}$.

**Shape parameter.** The shape parameter $\Gamma_p = \Gamma_p(\boldsymbol{r})$ is constructed upon averaging the shape function $\gamma_p$ of the segmented cells whose center of mass, $\boldsymbol{r}_c$, lies within a disk of radius $R$ centered at $\boldsymbol{r}$. That is

$$\Gamma_p(\boldsymbol{r}, R) = \frac{1}{N_{disk}}\sum_{c=1}^{N_{cell}}\gamma_p(\boldsymbol{r}_c)\Theta(R - |\boldsymbol{r} - \boldsymbol{r}_c|). \qquad (3)$$

Here $\Theta(x)$ is the Heaviside step function – such that $\Theta(x) = 1$ for $x > 0$ and $\Theta(x) = 0$ otherwise – and $N_{disk} = \sum_c\Theta(R - |\boldsymbol{r} - \boldsymbol{r}_c|)$, the number of cells whose centers lie within the disk and $N_{cell}$ the total number of cells.

The position $\boldsymbol{r}$ is sampled from a square grid with lattice spacing equal to half of the mean cell-cell distance $R_{cc}$, within a field of view of $124 \times 124\,\mu m$: i.e. $\boldsymbol{r} \in \{\boldsymbol{r}_n\}$ with $n = 1, 2\ldots N_{grid}$. To construct $\Gamma_p$ at increasingly larger length scales, the coarse-graining radius $R$ is increased by steps of $R_{cc}/2$. The scale-dependent shape parameter is finally averaged across each sample to obtain the data presented in Supplementary Fig. S1c, d. That is

$$\overline{|\Gamma_p(R)|} = \frac{1}{N_{grid}}\sum_{n=1}^{N_{grid}}|\Gamma_p(\boldsymbol{r}_n, R)|. \qquad (4)$$

We stress that, in Eq. (2), the summation runs over all the cells in our dataset, thus from different samples, whereas that in Eq. (4) is restricted to an individual sample. Furthermore, the sampling points $\{\boldsymbol{r}_n\}$ do not correspond the center of the cells.

**Topological defects.** Topological defects are identified by computing the winding number $s$ along the contour of a unit cell of the aforementioned square grid. That is

$$s = \frac{1}{2\pi}\oint_{\square}d\theta = \frac{1}{2\pi}\sum_{n=1}^{4}\left[\theta(\boldsymbol{r}_{n+1}) - \theta(\boldsymbol{r}_n)\right]\bmod\frac{2\pi}{p}, \qquad (5)$$

where the symbol $\square$ denotes a square unit cell in the interpolation grid and the mod operator constraint with $\theta = Arg(\Gamma_p)/p$ the phase of the coarse grained field $\Gamma_p$.

## Statistics
The number of experiments performed and number of images taken are summarized in Supplementary Table S1. All data sets are of non-normal distribution. $P$-values between two groups were calculated using the two-sided Wilcoxon rank sum test in MATLAB. The null hypothesis is fulfilled if the medians are equal. Comparisons between more than two groups were performed using Dunn's test of multiple comparisons after Kruskal-Wallis significance test in R. Data set significance was defined as of $p \le 0.05$ (*); $p < 0.01$ (**); $p < 0.001$ (***); $p < 0.0001$ (****); $p > 0.05$ (ns).

## Reporting summary
Further information on research design is available in the Nature Portfolio Reporting Summary linked to this article.

## Data availability
The raw data used in this study are available from Zenodo[43] [https://doi.org/10.5281/zenodo.8233583]. The data generated in this study are provided in the Source Data file. Source data are provided with this paper.

## Code availability
The MATLAB code is available on GitHub [https://github.com/hexanematic/orientation_tracker].

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

## Acknowledgements

We acknowledge the ImagoSeine core facility of the IJM, a member of IBiSA and France-BioImaging (ANR-10-INBS-04) infrastructures. J.E. acknowledges EMBO for a Scientific Exchange Grant (STF reference number: 9076) to financially support the research visit to Benoît Ladoux's lab at the Institut Jacques Monod in Paris. This work is part of the research program 'The Active Matter Physics of Collective Metastasis' (L.G. and T.S.), project number Science-XL 2019.022, which is financed by the Dutch Research Council (NWO). Further it is supported by the European Research Council (Grant No. Adv-101019835 to B.L.), LABEX Who Am I? (ANR-11-LABX-0071 to B.L. and R.M.M.), the Ligue Contre le Cancer (Equipe labellisée 2019 to R.M.M. and B.L.), the DIM "Elicit" Région Ile-de-France (B.L., R.M.M.), and the European Research Council (ERC) via the grant HexaTissue (L.G.). J.E. acknowledges Wang Xi for the stiffness measurement of the PAA gels, and Josep-Maria Armengol-Collado for the discussions on the theoretical aspects of this work.

## Author contributions

J.E. conducted and coordinated the research, performed the experiments, and analyzed the data. T.S., B.L., R.M.M., and L.G. supervised the project. All authors conceptualized the study, wrote the manuscript, and agreed on its current version.

## Competing interests

The authors declare no competing interests.
