## [Peer Review File · Nature Communications]

Hexanematic crossover in epithelial monolayers depends on cell adhesion and cell densityReviewer #1 (Remarks to the Author):

The manuscript submitted by Julia Eckert et al identifies a crossover scale between hexanematic order (rotational symmetry $p=6$) at small scales and nematic order (rotational symmetry $p=2$) at large scales. Built on recent theoretical approaches to tissues as liquid crystals, they explore the behaviour of Madin-Darby canine kidney cells (MDCK-II). The crossover scale is shown to be strongly affected by cell density but not by substrate stiffness. In addition, they show that the presence of hexanematic order relies on the E-cad-mediated intercellular adhesion, as they show experimentally that E-Cad KO cells tend to have elongated shapes and thereby the tendency of organizing in nematic phases.

Disentangling the multilevel organization of biological systems and identifying the characteristic scale levels is one of the key problems of biological physics. In that case, the existence of two different behaviours in the tiling of the space by cells in confluent tissues may have important consequences in tissue growth, migration patterns and wound healing. The conclusions of the paper are extracted through a combination of recent theoretical approaches and experimental results. Therefore, it touches relevant topics on the field and is timely. There are, nevertheless, several points that prevent me to recommend the publication in its current stage. When satisfactorily amended, I think the paper has the potential for being published in a general audience journal like Natcomms. I advance I don't know if I properly understood all the paper, so maybe some of my comments are out of place --I will be pleased if I can be proven wrong! Nevertheless, the authors should bear in mind that Natcomms is a journal with a broad audience and that their paper could have large visibility if properly explained.

My comments will be based on the statistics and mathematical framework:

1. The manuscript is, in its current form, somewhat difficult to follow. In order to improve the readability, I would propose that a theoretical scheme showing the main theoretical concepts is needed. In Fig. 1, for example, a scheme of what does it mean to ask for the order in the different rotational-symmetry based indexes is necessary. In addition, a scheme of what is a cross-over between hexanematic and 2-nematic order would be very useful. With that, the reader who is not specialist with the exact methods proposed in the manuscript can have an idea of what phenomenon is presented.

2. I guess in all the manuscript the assumption of confluence is at work, although this is sometimes not totally clear. In addition, increasing/decreasing density seems to be a key factor throughout all the discussions. Does it imply that cells can be compressed? On the contrary, reduction of the density may lead the tissue in tension. How does this affect the results? In a low density tissue, where, according to the authors, 2-nematic order seems to be more pronounced, can the authors rule out any external tension? Is it related to the surface stiffness? Please clarify this point.

3. Related to the above point, apparently the difference $\langle |\gamma_6| \rangle - \langle |\gamma_2| \rangle$ seems to be relevant. Can they be compared? Even normalized, one is based in the standard 2-norm for computing the modulus, and the other uses the 6-norm... Doesn't it introduce a by-default quantitative difference? I wonder how a null model would be expected behave --i.e., perfect (p)-nematic order, or complete disorder. Do they behave comparably? Maybe yes, and what I say is unfounded: In any case, and given that the (promising) theoretical framework is new, it would be great to show that the scalars of shape indexes have comparable behaviour.

4. The paper is built upon the theoretical analysis provided in refs. 30 and 31. In the manuscript the theoretical approach is poorly explained. Please enlarge the supplementary material such that the paper is better self-contained. Please, frame better the different elements involved. In addition, the current version seems to have some mistakes. In particular, in the supplementary material:

- Equation 1: State that "V" is the number of vertices of the 2D polygon
- Equation 1: State that "p" is the rotational symmetry considered
- Line 4 after equation 1: The definition of r_c has a mistake, as the maximum value of the sum index should be "V" instead of "N"

- Equation 2: Specify what the index c is --i.e., the c -th cell.
- Equation 3: Specify what γ_k means.
- Is N different from N_R ?

5. In the statistics, there is a constant use of the p-value. For example, in page 3, 1st column, line 24, it is stated that the mean \pm sd of the cell index has a p-value <0.00001 . Why do you need the p-value here? Considering that the sd is lower than an order of magnitude of the mean, I think there are no reasons to consider that the mean \pm sd is not valid enough? Against which null-model is the p-value computed? I may misunderstand something here so, please, clarify this point.

6. Ref. 30 seems to touch quite similar topics and reaches similar results, with some of the authors also being in the author list of the manuscript under revision. The presence of overlap is not a priori a problem, please clarify what is new in this submission.

7. There are some typos in the text, hence it should be thoroughly checked.

I may better appreciate the depth of the results provided in the paper if I read a revised version of it, which I will be pleased to do. Anyway, I feel the paper has potential enough to be published in Natcomms.

Reviewer #2 (Remarks to the Author):

Mechanisms of cell collective motion in biological tissues depend on the length scale on which motion is considered. Since different kinds of cell ordering prevail at different scales, hexatic at smaller and nematic at larger scales, it results that local order and collective motion mechanisms are strictly related. Tissues dynamics is very relevant in many biological processes, therefore it is important to study the behavior of the hexatic-nematic crossover length and its dependence on different system parameters. Hexatic and nematic order with its crossover are experimentally studied in this article for different systems. It is demonstrated that the hexanematic crossover shifts towards shorter length scales for decreasing monolayer density and reduction of the cell-cell interaction. This is a central result well demonstrated in this study that I believe noteworthy for the research community in this field and for more general biological implications. The experimental methodology and data analysis well support the conclusions presented. The paper is written in a very clear way. I recommend the publication of this work in its present form.

Manuscript Title: Hexanematic crossover in epithelial monolayers depends on cell adhesion and cell density

We thank both reviewers for taking the time to review the manuscript and for their insightful comments. We have incorporated changes in the manuscript to address the suggestions made by the reviewers. Responses to the reviewers' comments can be found point-by-point in the text below. All changes in the manuscript are highlighted red.

Reply to Reviewer #1

[...] There are, nevertheless, several points that prevent me to recommend the publication in its current stage. When satisfactorily amended, I think the paper has the potential for being published in a general audience journal like Natcomms. I advance I don't know if I properly understood all the paper, so maybe some of my comments are out of place --I will be pleased if I can be proven wrong! Nevertheless, the authors should bear in mind that Natcomms is a journal with a broad audience and that their paper could have large visibility if properly explained.

Reply: We thank Reviewer #1 for communicating the concerns regarding the clarity of our manuscript. To improve we revised parts of the content based on the comments raised. We hope that the improved version further clarifies our message, and by this can be appreciated by a broader audience. See the text in red for our changes.

1. The manuscript is, in its current form, somewhat difficult to follow. In order to improve the readability, I would propose that a theoretical scheme showing the main theoretical concepts is needed. In Fig. 1, for example, a scheme of what does it mean to ask for the order in the different rotational-symmetry based indexes is necessary. In addition, a scheme of what is a cross-over between hexanematic and 2-nematic order would be very useful. With that, the reader who is not specialist with the exact methods proposed in the manuscript can have an idea of what phenomenon is presented.

Reply: The revised version of the manuscript includes a whole new figure, Fig. 1, where we have illustrated the concept of multiscale order as well as that of hexanematic crossover at a more intuitive level. In addition to this, we have substantially expanded the introduction, to explain the relevance of this fascinating form of spatial organisation in tissues, and the methods section and Fig. 3 (previously Fig. 2), to provide the readers with a precise and pedagogical account of how the various quantities are computed.

2. I guess in all the manuscript the assumption of confluence is at work, although this is sometimes not totally clear. In addition, increasing/decreasing density seems to be a key factor throughout all the discussions. Does it imply that cells can be compressed? On the contrary, reduction of the density may lead the tissue in tension. How does this affect the results? In a low density tissue, where, according to the authors, 2-nematic order seems to be more pronounced, can the authors rule out any external tension? Is it related to the surface stiffness? Please clarify this point.

Reply: Experimentally, we seeded cells at different starting concentrations on coverslips. After seeding, epithelial cells will (within limits) cover all area available to them. Thus, different seeding concentrations naturally result in different cell densities (cells per area). Cells at lower densities tend to be more elongated than cells at higher densities, see Refs (19) and (42) in the manuscript. These density variations affect, for example, the cortical tension and traction forces of cells. In our manuscript, we capture the impact of density variations on the tissue's symmetry.

As Reviewer #1 notices, varying the cell density does affect the stress distribution of the monolayer. However, as the latter is not laterally confined, such an effect is not the same that would result from an externally applied lateral compression or extension of the entire cell monolayer. Conversely, all stresses at play in the monolayer are internally generated, with only exception for the hydrostatic pressure resulting from the fluid in which the monolayer is embedded. Being stress an extensive quantity, the larger the number of cells pulling (as a result of contractility) and pushing (as a result of motility) on each other, the larger the local stress. Now, as some of us theoretically predicted in Ref. [31], increasing the active stress at the cellular scale results in an increase of the crossover scale R_c , thereby extending the range of length scales at which hexatic order prevails over nematic order. Our observations confirms this predictions. It should be noticed that, whether at low or high density, our cell monolayers never experience a *net* extensile stress, as this would result in breakdown of confluency, which, instead, it is never observed. By contrast, and consistently with Ref. [41] (Saraswathibhatla and Notbohm, *Phys. Rev. X* 2020), increasing cell density decreases the alignment among stress fibers, thus further strengthening hexatic order. Analogously, decreasing density increases the alignment of the stress fibers, thereby enhancing the unidirectional stresses form which nematic order originates. We have elaborated on these concepts at the end of the section “Hexatic order strengthen with the monolayer density”.

Finally, as Reviewer #1 suggests, cellular contractility is counterbalanced, at least in part, by the substrate. For this reason experiments were performed for a range of substrate stiffnesses (see “Lower substrate stiffnesses reinforce the length scale of the hexatic order driven by cell-matrix and cell-cell adhesions”). Our findings indicate that unidirectional cellular contractility increases with substrate stiffness, thus leading to the shift of the hexanematic crossover towards smaller length scales.

3. Related to the above point, apparently the difference $\langle |\gamma_6| \rangle - \langle |\gamma_2| \rangle$ seems to be relevant. Can they be compared? Even normalized, one is based in the standard 2-norm for computing the modulus, and the other uses the 6-norm... Doesn't it introduce a by-default quantitative difference? I wonder how a null model would be expected behave --i.e., perfect (p)-nematic order, or complete disorder. Do they behave comparably? Maybe yes, and what I say is unfounded: In any case, and given that the (promising) theoretical framework is new, it would be great to show that the scalars of shape indexes have comparable behaviour.

Reply: Regardless of the specific value of the integer p , all shape functions γ_p , as defined in Eq. (1), are dimensionless complex numbers. As explained in the section “Hexagonality of epithelia increases with density”, the magnitude of these complex number expresses the resemblance of a cell to a regular p -sided polygon having the same position and size. The phase, on the other hand, indicates the p -fold orientation of a cell. As Reviewer #1 observes, a comparison between different p -fold orientations requires special care and is not always meaningful, as in epithelial layers nematic and hexatic order inhabit different length scales. On the other hand, the difference $\langle |\gamma_6| \rangle - \langle |\gamma_2| \rangle$ between the *magnitudes* of 6-fold and 2-fold shape functions at the scale of individual cells is always well defined and quantifies whether cells are in average closer in shape to hexagons (when positive) or rods (when negative).

4. The paper is built upon the theoretical analysis provided in refs. 30 and 31. In the manuscript the theoretical approach is poorly explained. Please enlarge the supplementay material such that the paper is better self-contained. Please, frame better the different elements involved. In addition, the current version seems to have some mistakes. In particular, in the supplementary material:

·Equation 1: State that “ V ” is the number of vertices of the 2D polygon

·Equation 1: State that “ p ” is the rotational symemtry considered

- Line 4 after equation 1: The definition of r_c has a mistake, as the maximum value of the sum index should be “ V ” instead of “ N ”
- Equation 2: Specify what the index $_c$ is --i.e., the c -th cell.
- Equation 3: Specify what $\gamma_{k,k}$ means.
- Is N different from N_R ?

Reply: Thank you for reading our methods section in detail and pointing out some mistakes. We have corrected these points in our manuscript, and extended the sections with more explanations.

5. In the statistics, there is a constant use of the p-value. For example, in page 3, 1st column, line 24, it is stated that the mean \pm sd of the cell index has a p-value <0.00001 . Why do you need the p-value here? Considering that the sd is lower than an order of magnitude of the mean, I think there are no reasons to consider that the mean \pm sd is not valid enough? Against which null-model is the p-value computed? I may misunderstand something here so, please, clarify this point.

Reply: The sentence mentioned refers to: “MDCK WT cells had a smaller shape index of 4.06 ± 0.07 (mean \pm s.d.) compared to MDCK E-cad KO cells with 4.20 ± 0.21 (mean \pm s.d.), p-value < 0.0001 ”. We consider the p-value calculation as a standard procedure when comparing results of two independent data sets. To calculate the p-value, we used the two-sided Wilcoxon rank sum test. This test compares the distribution of two independent data sets. The null hypothesis is fulfilled if the medians are equal, we have added it to the supplementary. In this particular case, it is obvious that the null hypothesis is rejected because the medians are far apart, as shown in Fig.1C. Nevertheless, the mean shape index of 4.06 of MDCK WT cells is within the error range of 4.41 ($=4.2+0.21$) of the MDCK E-cad KO cells. To prove a significant difference between the two values, we calculated the p-value.

6. Ref. [30] seems to touch quite similar topics and reaches similar results, with some of the authors also being in the author list of the manuscript under revision. The presence of overlap is not a priori a problem, please clarify what is new in this submission.

Reply: In Ref. [30] two of us introduced a number of mathematical tools, such as the formation shape function γ_p and its coarse-grained version — i.e. the shape parameter Γ_p — meant at characterizing multiscale hexanematic order, whose existence was predicted in Ref. [31], in confluent cell layers. In this manuscript we used these tools to expand the current understanding of multiscale hexanematic order to include aspects which are hardly accessible to both continuum and discrete theories of tissues, especially those that, like in Ref. [31], focus on the mechanical aspects of collective cell behavior. These includes, in particular, the role of adhesion of the cells between each other and with the substrate, which, being mediated by a complex regulatory network of signaling pathways, is *de facto* impossible to be accurately incorporated in a mechanical model. We have added a comment on this at the end of the introduction.

7. There are some typos in the text, hence it should be thoroughly checked. I may better appreciate the depth of the results provided in the paper if I read a revised version of it, which I will be pleased to do. Anyway, I feel the paper has potential enough to be published in Natcomms.

Reply: We have corrected the typos in the manuscript and hope that we found all of them.

Reviewer #1 (Remarks to the Author):

The revised version of the manuscript "Hexanematic crossover in epithelial monolayers depends on cell adhesion and cell density" addressed successfully comments. The responses of the authors to my comments were also satisfactory.

I spotted a small typo in equation (2): It should read N_{cell} instead of N_{cells} , if one wants to be in agreement with the definitions.

I thereby recommend the manuscript for publication.

With best regards